# Genomic Insights into Moderately Thermophilic Methanotrophs of the Genus *Methylocaldum*

**DOI:** 10.3390/microorganisms12030469

**Published:** 2024-02-26

**Authors:** Nathalie A. Delherbe, David Pearce, Sergey Y. But, J. Colin Murrell, Valentina N. Khmelenina, Marina G. Kalyuzhnaya

**Affiliations:** 1Department of Biology, San Diego State University, San Diego, CA 92129, USA; ndelherbe@sdsu.edu; 2School of Environmental Sciences, University of East Anglia, Norwich NR4 7TJ, UKj.c.murrell@uea.ac.uk (J.C.M.); 3G. K. Skryabin Institute of Biochemistry and Physiology of Microorganisms, Scientific Center for Biological Research of the Russian Academy of Sciences, 142290 Pushchino, Russia; flash20063@rambler.ru (S.Y.B.); khmelenina@rambler.ru (V.N.K.)

**Keywords:** thermophilic methanotrophic bacteria, *Methylocaldum*, methane monooxygenase, pyomelanin, methanol dehydrogenase, extremophiles

## Abstract

Considering the increasing interest in understanding the biotic component of methane removal from our atmosphere, it becomes essential to study the physiological characteristics and genomic potential of methanotroph isolates, especially their traits allowing them to adapt to elevated growth temperatures. The genetic signatures of *Methylocaldum* species have been detected in many terrestrial and aquatic ecosystems. A small set of representatives of this genus has been isolated and maintained in culture. The genus is commonly described as moderately thermophilic, with the growth optimum reaching 50 °C for some strains. Here, we present a comparative analysis of genomes of three *Methylocaldum* strains—two terrestrial *M. szegediense* strains (O-12 and Norfolk) and one marine strain, *Methylocaldum marinum* (S8). The examination of the core genome inventory of this genus uncovers significant redundancy in primary metabolic pathways, including the machinery for methane oxidation (numerous copies of *pmo* genes) and methanol oxidation (duplications of *mxa*F, *xox*F1-5 genes), three pathways for one-carbon (C1) assimilation, and two methods of carbon storage (glycogen and polyhydroxyalkanoates). We also investigate the genetics of melanin production pathways as a key feature of the genus.

## 1. Introduction

Microbial methane oxidation is a key process in the carbon cycle at local and global scales. Methane-utilizing bacteria (methanotrophs) inhabiting high-temperature ecosystems include members of the genera *Methylococcus*, *Methylothermus*, and *Methylocaldum* from the phylum Pseudomonadota (Proteobacteria) and the genus *Methylacidiphilum* in the phylum Verrucomicrobia [1,2,3]. Thermophilic and thermotolerant species of the *Methylocaldum* genus (Gammaproteobacteria class in the family *Methylococcales*) are commonly identified as important members of the bacterial communities in soils from a range of geographical locations [4], including hot springs [5], landfill cover soils [6,7], tin-mining ponds [8], oil sands [9], flooded rice fields [10], and marine sediments [11]. The genus *Methylocaldum* was introduced by Bodrossy et al. (1997) [5]. The genus currently includes four species: *M. szegediense*, *M. tepidum*, *M. gracile* [5], and *M. marinum* [11]. The isolate *M. szegediense* O-12 was obtained from a sample of cow manure from a farm near Pushchino in the Moscow region in Russia. The pure culture was obtained from an initial enrichment, followed by a serial dilution to extinction technique [12]. *M. szegediense* O-12 grows at a temperature range of 37–59 °C, with optimal growth at 55 °C. *M. szegediense* Norfolk was isolated from biofilter soil enrichments from a landfill in Norfolk County Council, UK, and grows between 37 °C and 62 °C, with an optimal growth at 50 °C [13]. Here we conduct a comparative analysis between thermotolerant methanotrophs to identify key genes potentially involved in their capability to grow at elevated temperatures. The genomic features of the two terrestrial, moderate thermophilic *M. szegediense* O-12 and Norfolk strains are compared with the thermotolerant *M. marinus* S8 (=NBRC 109686^T^ = DSM 27392^T^) isolated from marine sediments in Kagoshima Bay, Japan, which grows optimally at 36 °C and a salinity of up to 5% NaCl [11].

## 2. Materials and Methods

**Genome sequencing, assembly, and annotation.** *M*. *szegediense* O-12 (taxon ID: 2508501066 (IMG)/675524 (NCBI)) genome sequencing and assembly was conducted in collaboration with U.S. Department of Energy Joint Genome Institute (NCBI assembly ASM42738v1). The genome and assembly of *M. szegediense* Norfolk (taxon ID: 8069803063 (IMG)/73780 (NCBI)) was generated by the laboratory of Dr. J. Colin Murrell, at University of East Anglia, Norwich, UK, utilizing an enhanced genome sequencing coupling short Illumina and long Nanopore reads (NCBI assembly strumpshaw02) [13]. The genome sequence of *M. marinum* S8 (taxon ID: 2832923104 (IMG)/1432792 (NCBI)) was generated by National Institute of Advanced Industrial Science and Technology (AIST), Japan (NCBI assembly ASM358464v1). In addition to being available in NCBI, the three genomes were uploaded and annotated in IMG/MER [14,15].

**Methylocaldum phylogenic analysis** was carried out using all (20) complete sequences available for the 16S rRNA gene assigned to *Methylocaldum* strains on NCBI at the time of the analysis (2022). The alignment was generated using the multiple alignment program for amino acid or nucleotide sequences MAFFT v7.511, choosing the E-INS-i method [16,17,18]. The phylogenetic relationship was inferred using the MEGA X software v10.2.6 using the maximum likelihood method and the model with the lowest Bayesian information criterion (BIC) scores for the set, which corresponded to Tamura 3-parameter (T92) with a discrete Gamma distribution (+G) with 5 rate categories and by assuming that a certain fraction of sites is evolutionarily invariable (+I) [19,20,21,22]. The analysis involved 32 nucleotide sequences (including *Methylococcus capsulatus* Bath and *Methylobacter* species as outgroups), and all positions with less than 95% site coverage were eliminated.

**pMMO synteny, functional genomic comparison, and ANI x AC.** The synteny and functional genomic comparison of the genomes was based on the information obtained using Anvi’o v7 software [23]. The visual representation of the pMMO operon synteny was generated by combining the outputs of Circa plot (v.1.2.1) utilizing the corresponding nucleotidic sequence coordinate data and combining them with the gene cluster visualization output from Geneious software v.2023.2.1. The generated data included a list of all coding sequences (CDS) from the detected open reading frames identified by Prodigal v.2.6.3 [24]. The CDS predicted functionality was annotated using the NCBI’s Clusters of Orthologous Groups (COG) database [25,26,27] and the KEGG KOfamKOALA database [28]. The pangenome analysis, based on the pangenome pipeline of Anvi’o v7 [29], generated orthologous groups (OG) of genes based on the homology of their amino acid sequences. A >70% of combined homogeneity index (derived from geometrical and functional homology) was the cut-off to consider a gene as part of an OG. Each OG contained at least one gene. Additionally, the OGs were organized by presence/absence through Euclidean distance and Ward linkage, which was used to categorize them as *shared* by the three genomes or *unique* for each of them. Then, within each of the shared and unique subsets, all CDS with COG assigned to them were used to determine which COG categories were the most represented (>5%). In addition to the top of the pangenome analysis, a whole genome comparison was generated using the program *anvi-compute-genome-similarity* utilizing the PyANI [22] similarity metric to calculate the average nucleotide identity (ANI). Additionally, the full percent identity was calculated for the ANI, which includes the regions that did not align, by multiplying the percent identity by the alignment coverage (ANI x AC).

***pmoC* homologs**. Genomes of *Methylocaldum*, *Methylobacter*, and *Methylococcus* available on IMG/MER were used to search for the range of copies of *pmo*C. The search was based on IMG annotations using the different available identifiers for *pmo*C: TIGR03078, KO10946, and gene product name. A table containing the Anvi’o v7 results for synteny and functional genomic comparison of the genomes can be found in Appendix A.

**XoxF phylogeny.** The alignment was generated using the multiple alignment program for amino acid or nucleotide sequences v7.511 [17,18], choosing the E-INS-i method [16,17,18]. The tree topology was obtained after 100 bootstrap replications using the maximum likelihood method and general time reversible model [20] using MEGA X software v10.2.6 [19,21]. The tree with the highest log likelihood (−175,992.24) [22] was used to generate the summarized version of it.

**Fdh phylogeny.** The alignment was generated using the multiple alignment program for amino acid or nucleotide sequences MAFFT v7.511 [17,18], choosing the E-INS-i method [16,17,18]. The tree topology was obtained after 100 bootstrap replications using the maximum likelihood method and the Whelan and Goldman + Freq. model [30]. The tree with the highest log likelihood (−50,331.00) (Appendix A) was used to generate the cartoon version of it. The percentage of trees in which the associated taxa clustered together is shown next to the branches. This analysis of 193 amino acid sequences was conducted in MEGA X v10.2.6 [19,21]. There were a total of 1638 positions in the final dataset.

## 3. Results

### 3.1. Genome Overview

The *M. szegediense* O-12 genome (taxon ID: 2508501066 (IMG)/675524 (NCBI)) includes two scaffolds of 2.37 Mb and 2.65 Mb, resulting in a genome of 5.02 Mb with an average GC content of 57.03%. It includes two rRNA operons, 22 types of tRNAs (47 total), and 4693 predicted protein-coding genes (3448 predicted functions). The genome of *M. szegediense* sp. Norfolk (taxon ID: 8069803063 (IMG)/73780 (NCBI)) includes a closed genome comprising one chromosome of 4.87 Mb and a 25.7 kb plasmid, resulting in a genome of 4.89 Mb with an average GC content of 57.17%. It contains two rRNA operons, 24 types of tRNAs (53 total), and 4491 predicted protein-coding genes (3378 predicted functions). *Methylocaldum marinum* S8 (taxon ID: 2832923104 (IMG)/1432792 (NCBI)) has a complete genome of 6.08 Mb with a 58.75% GC content, including two rRNA operons, 24 types of tRNAs (51 total), and 5596 predicted protein-coding genes (3846 predicted functions).

### 3.2. Methylocaldum Phylogeny

The phylogeny of *Methylocaldum* strains based on available 16S rRNA sequences shows five clades (Figure 1A). Four of these clades correspond to currently characterized species within the *Methylocaldum* genera (*Methylocaldum tepidum* LK6^T^, *Methylocaldum szegediense* OR2^T^, *Methylocaldum gracile* VKM 14L^T^, and *Methylocaldum marinum* S8^T^), while the fifth clade formed by strains BFH1, dr65, and r6f does not have a type strain. A comparison based on the presence or absence of orthologous genes between the three strains analyzed in this study revealed that the *M. szegediense* O-12 and Norfolk strains share 90% and 93%, respectively, of their predicted protein sequences as orthologous, while only 47% of the predicted protein sequences in *M. marinum* S8 were orthologous to genes in the *M. szegediense* O-12 and Norfolk strains (Figure 1B). The average nucleotide identity times the alignment coverage (ANI x AC) percentage between the three strains confirmed that *M. szegediense* O-12 and Norfolk are members of the same species (≥90%), while *M. marinum* S8 shared only 38% with the *M. szegediense* strains (Figure 1B).

### 3.3. Synteny of Methane Monooxygenase (MMO) Gene Clusters

The identification of key genes necessary for methane oxidation in bacteria revealed the presence of several copies of the *pmo*C gene, which is usually part of the *pmo*CAB operon for the particulate methane monooxygenase (pMMO). Five copies were found in *M. marinum* S8, six were found in *M. szegediense* O-12, and five were found in *M. szegediense* Norfolk (Figure 2). In the *M. marinum* S8 and *M. szegediense* Norfolk strains, two *pmo*C genes formed part of the complete gene cluster for the synthesis of the pMMO, whereas only one *pmo*C gene formed part of the complete gene cluster in *M. szegediense* O-12 (Figure 2). All *pmo*C genes forming part of pMMO operons had high sequence identity to each other and were denominated as type 1 (purple ribbon, Figure 2). Furthermore, the three *Methylocaldum* genomes contained paralogs of *pmo*C as independent genetic components in different loci of their genomes, which have been previously named as stand-alone copies of *pmo*C [31]. Except for a single stand-alone *pmo*C in *M. szegediense* O-12 assigned to type 1, the rest of the stand-alone copies of *pmo*C have different nucleotide sequences than the one forming the pMMO cluster (type 1) and form two separate groups of paralogs, denominated as type 2 and 5 (pink and magenta ribbons, respectively, Figure 2). Furthermore, both *M. szegediense* strains exhibit two stand-alone *pmoC* type 2 relatively next to each other. Additionally, two stand-alone *pmo*C without high homology to any other *pmoC* sequences within the three genomes were found in *M. marinum* S8 (type 4) and *M. szegediense* O-12 (type 3), shown in black-edged triangles in Figure 2. Only *M. marinum* S8 exhibits a complete gene cluster for the soluble methane monooxygenase (sMMO). When compared with other methanotrophs with genomes annotated on IMG/MER (*Methylobacter* [16] and *Methylococcus* [10]), *Methylocaldum* strains have between five and six coding sequences (CDS) assigned to *pmo*C, while *Methylobacter* strains have between one and two and *Methylococcus* have between two and three *pmo*C in the available genomes.

### 3.4. Comparative Abundance CDS Assigned to COG Categories

According to the pangenome analysis of the three *Methylocaldum* strains, from a total of 14,284 CDS predicted for the three genomes, 2214 CDS formed orthologous groups (OG) of amino acid sequences based on a ≥70% threshold of combined homogeneity index (derived from geometrical and functional homology; Figure 3). Each genome had at least one gene copy as part of each OG assigned as shared among the three strains. Based on their relative abundances, the main shared COG categories (comprising >5% of the total gene calls that could be categorized) were genes related to categories C (8.58%) involved in energy production and conversion; E (7.36%) amino acid transport and metabolism; H (5.74%) coenzyme transport and metabolism; J (8.04%) translation, ribosomal structure and biogenesis; M (7.09%) cell wall/membrane biogenesis; O (6.23%) post-translational modification; P (6.32%) inorganic ion transport and metabolism; and R (5.42%) classified only as general function.

On the contrary, the CDS found to be unique to each genome, based on a low combined homogeneity index of their amino acidic sequences when compared to their counterparts in the other genomes, were mainly assigned to COG categories M, cell wall/membrane biogenesis (11.57%); X, mobilome including prophages and transposons (10.9%); T, signal transduction mechanisms (8.15%); P, inorganic ion transport and metabolism (7.70%); R, general function (7.47%); V, defense mechanisms (5.67%); H, coenzyme transport and metabolism (5.17%); and C, energy production and conversion (5%) for *M. marinum* S8. At the same time, the pattern was similar between *M. szegediense* O-12 and Norfolk with V (13.91% and 23.07%), X (15.65% and 7.69%), L, replication, recombination, repair (17.39% and 16.92%), R (8.7% and 13.84), K, and transcription (8.70% and 13.84%), respectively (Figure 3).

### 3.5. Conjugation

Several predicted functions were only found in the two moderate thermophilic *M. szegediense* strains, including K12056, corresponding to the conjugal transfer mating pair stabilization protein TraG, which is one of the pilus assembly proteins in bacterial type IV secretion systems (T4SS) [32]. While *M. szegediense* Norfolk has only two conjugal transfer proteins, TraG and TraN, the *M. szegediense* O-12 genome contains two TraG, as well as TraW, TraF, TraE, and TraL, which participate in pilus assembly; TraA (pilin); TraN and TraU, which are responsible for mating pair stabilization; TraV, TraK, and TraB, which form the core complex; TraD and TraC, which are ATPase proteins reported to be complexed in the cytosol and inner membrane; and also TrbL, which is a subset of proteins described as F-like T4SS [32,33].

### 3.6. O-Antigen Biosynthesis

Interestingly, two homologs of O-antigen biosynthesis protein RfbC (K20444; COG0438|COG1216) were found only in the *M. szegediense* strains. RfbC is predicted to catalyze the incorporation of sugars and their products to form the O-antigen polysaccharides in the lipopolysaccharides (LPS) [34]. The enzyme’s specificity toward rare sugars enables the production of unusual LPS, thus providing the remarkable diversity of cell envelop recognition patterns observed among bacteria [35,36,37,38,39]. RfbC is a member of the glycosyltransferase family 2 (GT2). A search for other glycosyltransferases belonging to that family revealed the presence of six other OGs shared between the *M. szegediense* strains and absent in *M. marinum* S8. Additionally, the three *Methylocaldum* strains share three OGs of GT2, and *M. marinum* S8 has four different GT2, which are not shared with the *M. szegediense* strains. Differences between the genetic toolkit for the biosynthesis of the O-antigen among *Methylocaldum* offer a starting point for future studies to elucidate the mechanisms of specialization in their different niches and how they can tolerate a wide range of temperatures. It has been shown that O-antigen structural differences can confer resistance to different types of stress, such as oxidative [40], mechanical [39], and osmotic stresses [41]. Additionally, antigen molecular diversity is a current target of study to understand interdomain symbiosis [42].

### 3.7. Catalase

Among the distinct functions that were found to be unique for *M. marinum* S8 is catalase KatE (COG 0753, KEGG K03781), which can convert toxic hydrogen peroxide (H_2_O_2_) into water and oxygen. H_2_O_2_ is one of the three products of methanethiol (CH_3_SH) oxidation (HCOH, H_2_S, H_2_O_2_), for which all three genomes have a homolog to the gene *mtoX*, a copper-dependent methanethiol oxidase (MTO) gene characterized in *Hyphomicrobium* sp. VS [43] and annotated in our analysis as K17285. CH_3_SH is a volatile organic sulfur compound that can co-occur with methane. It can be consumed by the verrucomicrobial methanotroph *Methylacidiphilum fumariolicum* SolV, which produces H_2_S that is subsequently oxidized for energy [44]. It seems likely that this gene would be beneficial for methanotrophs since it could allow them to flourish in niches that release methane and CH_3_SH, an inhibitor of methane oxidation [45].

### 3.8. Cobalamin

Another unique function of *M. marinum* S8 among the three genomes was a nearly complete de novo cobalamin (B_12_) biosynthetic pathway. The only missing gene corresponds to *cbi*J-*cob*K, whose product, precorrin-6X reductase, converts (Co)Precorrin-6A to (Co)Precorrin-6B. This gene had been previously described as missing in almost all characterized cobalamin-producing archaea [46,47,48]. This suggests the independence of an external source of the essential cofactor, which would be an advantage, given that most predicted cobalamin producers in the ocean belong to photoautotrophic Cyanobacteria, chemoautotrophic Nitrososphaerota (Thaumarcheota), and a selected group of Pseudomonadota (Proteobacteria), including Rhodobacterales, Rhodospirillales, Deltaproteobacteria, Oceanospirillales, and Pseudomonadales [49]. Both *M. szegediense* strains lacked any recognizable genetic elements required for the corrinoid ring biosynthesis [50]. Nonetheless, all three genomes contain the genes for 5,6-dimethylbenzimidazole (DMB) synthesis, 5,6-dimethylbenzimidazole synthase; *blu*B (COG0778, K04719), which is the lower ligand of coenzyme B_12_; all the genes for the final synthesis and repair (K19222 (*cobA*), K00798 (*cobO*), K02232 (*cobQ*), K02225 (*cobC*), K02227 (*cobD*), K02231 (*cobP/cobU*), and K02233 (*cobS*); and, for B_12_ transport, K16092 (*btuB*), which match the previous classification of *M. szegediense* O-12 as potentially unable to do de novo synthesis, but instead to salvage cobamide precursors, assemble the nucleotide loop, and attach a lower ligand, based on its in silico analysis of 11,000 strains [51].

### 3.9. M. marinum S8 Unique Predicted Functions

Additionally, only *M. marinum* S8 has several hydrogenases subunits, including Fe-S-cluster-containing hydrogenase component 2 (HycB) COG1142, hydrogenase-4 membrane subunit HyfE COG4237 (K12140), Ni,Fe-hydrogenase III small subunit (HycG) (PDB:6CFW) COG3260, and the Ni,Fe-hydrogenase III large subunit HycE2|Ni,Fe-hydrogenase III component G (HycE1) COG3261|COG3262, which suggests the metabolic potential to obtain energy through membrane-associated H_2_ involving hydrogenases. These hydrogenases are in the immediate vicinity of homologs to the formate hydrogenlyase subunit HyfC (COG0650).

Other functions found only in the genome of *M. marinum* S8 are homologs for molybdopterin biosynthesis (COG0303|1910, K07219); peroxiredoxin DsrE/DsrF-like family (COG2044); putative NADH-Flavin reductase YWNB (COG2910); RecA superfamily ATPase KaiC/GvpD/Rad55 (COG0467, K0882); Fe^3+^ permease component FepD, (COG0609|0614) Fe^2+^ transporter FeoB|FeoA (COG0370|1918, K04759); periplasmic components for ABC-type tungstate transport system, TupA (COG4662, K05773) and TupB (COG2998, K05772); heavy metal-binding TRASH/YHS Cu/Ag metallochaperone (COG3350, K16157); nitrous oxide reductase accessory protein NosL (COG4314, K19342); and cobalt transporter membrane subunit CgtA (COG5446).

### 3.10. C1-Oxidation Pathways Are Highly Redundant

As described above, in addition to having two *pmo*CAB operons in the *M. szegediense* Norfolk and *M. marinum* S8 genomes and the sMMO operon in *M. marinum* S8, the three genomes also contain multiple stand-alone *pmo*Cs with unknown functions. They also have multiple methanol dehydrogenases. The *mxaFJGIRACKLD* gene clusters in the three *Methylocaldum* strains encode the two subunits of methanol dehydrogenase *mxa*F (K14028) and *mxa*I (K14029) and the natural electron acceptor, cytochrome c_L_ (mxaG). Moreover, the three strains share two OGs of additional methanol dehydrogenases, lanthanide-dependent MDHs, corresponding to *xoxF* types 5 and 3 according to the phylogenetic reconstructions (Figure 4) following previously *xoxF* type assignation [52,53,54]. *M. marinum* has a third *xoxF* with no homology to genes in the *M. szegediense* strains that also belongs to *xoxF* type 3.

The three *Methylocaldum* have the set of genes for the tetrahydromethanopterin (HMPT)-dependent pathway required for the oxidation of formaldehyde to formate. Gene redundancy was also found in this pathway. Based on the OG analysis, three different OGs for *fae* (K10713), which encodes the 5,6,7,8-tetrahydromethanopterin hydrolyase, were found among the *Methylocaldum*. One *fae* OG included three CDS from *M. marinum* S8, and two were included for *M. szegediense* O-12 and Norfolk. The other two OGs had only one CDS in each strain. Each strain had two methylene-tetrahydromethanopterin dehydrogenases *mtd*B (K10714) in separate OGs. Only one of the following genes was found in each of the strains: methylenetetrahydrofolate dehydrogenase *mtd*A (K00300), methenyltetrahydromethanopterin cyclohydrolase *mch* (K01499), formylmethanofuran--tetrahydromethanopterin N-formyltransferase *ftr* (K00672), formylmethanofuran dehydrogenase fwdABC (K00200, K00201, K00202), glycine hydroxymethyltransferase glyA (K00600), and coenzyme F420 hydrogenase subunit beta *frh*B (K00441), for which only *M. marinum* S8 had two CDS.

Interestingly, several formate dehydrogenases encoding genes (*fdh*) were found in the three strains. These enzymes catalyze the oxidation of formate to CO_2_, donating electrons to NAD^+^ or cytochromes. The phylogenetic reconstruction based on the amino acid sequence of the major subunits (K00123) of the different formate dehydrogenases found in *Methylocaldum* revealed that they have four types of *fdh* (Figure 5 and Appendix A). According to the previously described types of *fdh* in *Methylobacterium extorquens* AM1 [55,56], we detected the presence of three different *fdh* in the three *Methylocaldum* strains, corresponding to *fdh* types 2 and 3, including all their subunits. Additionally, FDH (K00122) was found in the three genomes as well. Moreover, *M. marinum* S8 has the major subunit for *fdh* type 1, although it was previously described to require the subunit B for functioning in the *M. extorquens* AM1 model [56,57].

### 3.11. Single-Carbon Assimilation Pathways

These *Methylocaldum* genomes encode three single-carbon assimilation pathways—the ribulose monophosphate (RuMP), serine, and Calvin–Benson–Bassham (CBB) cycle. The genes present in the RuMP pathway include two CDS for 3-hexulose-6-phosphate synthase in each genome, *hxl*A (K08093) and *hps-pi* (K13831), catalyzing the condensation of formaldehyde with ribulose-5-phosphate to D-arabino-hex-3-ulose 6-phosphate, which is converted to D-fructose 6-phosphate by the 6-phospho-3-hexuloisomerase, *hxl*B (K08094) present in the three genomes. D-fructose 6-P can then enter the pentose phosphate pathway.

The large *rbc*L/*cbb*L (K01601) and small *rbc*S/*cbb*S (K01602) subunits of the ribulose 1,5-bisphosphate carboxylase/oxygenase (RuBisCO) were found in the three strains, having a single gene difference between *M. szegediense* and *M. marinum*. This includes the presence of *nor*Q and *nor*D next to RuBisCO genes of the three *Methylocaldum*. *M. marinum* S8 also has pimeloyl-ACP methyl ester carboxylesterase between the RuBisCO and *nor* genes. The presence of *rcb*/*cbb* genes had been reported in other methanotrophs; however, its activity had not been demonstrated yet [58,59,60].

Similarly to other gammaproteobacterial methanotrophs, the regeneration of pentose phosphates from hexosephosphates requires fructose-bisphosphate aldolase, for which class I, ALDO (K01623), is present in all three *Methylocaldum* genomes.

As in other gammaproteobacterial methanotrophs, the *Methylocaldum* genomes contain some key genes for the serine pathway of formaldehyde assimilation. These include serine-glyoxylate transaminase, AGXT (K00830), which converts glyoxylate to hydroxypyruvate and glycine hydroxymethyltransferase, *gly*A (K00600), which uses 5,10-methylenetetrahydrofolate and glycine as substrates to produce tetrahydrofolate and L-serine, which can continue to the carbon fixation pathway or sulfur or serine metabolism, respectively. Also, as part of the serine pathway, the three genomes encode malyl-CoA/(S)-citramalyl-CoA lyase *mcl* (K08691), which produces acetyl-CoA and glyoxylate from L-malyl-CoA as a substrate. However, all strains lack phosphoenolpyruvate carboxylase, *ppc* (K01595), which produces oxaloacetate from phosphoenolpyruvate, necessary for the serine pathway. Anaplerotic functions of the serine pathway have been previously proposed in *Methylococcus capsulatus* Bath. Like the strain Bath, all three genomes have the necessary genes to convert malyl-CoA (*mcl*, K08691) to glyoxylate, then to glycine (AGXT, K00830), and subsequently to serine (*gly*A, K00600) as part of the serine pathway. In the same pathway, they also have the genes to convert from D-glycerate to glycerate-2P (*gck*, K11529) to phosphoenolpyruvate (*eno*, K01689).

### 3.12. Nitrogen Metabolism

All three strains of *Methylocaldum* have the structural genes for nitrogenase (*nif*H (K02588), *nif*D (K02586), and *nif*K (K02591)) in an operon with *nif*E, *nif*N, *nif*X, and *nif*Q genes and also a *nif*-specific ferredoxin III (TIGR02936) (Figure 6B). Two additional homologs of *nifH* and *nifK* genes were also found in the three strains, although their genetic context was different from each other and may not be related to nitrogen fixation. The genetic elements required for nitrate assimilation were found in the three strains, having as a difference the additional presence of the *nrt*ABC gene cluster (K15576, K15577, K15578) for the nitrate/nitrite transport system in *M. marinum* S8 only (Figure 6A). The three strains have the *nor*CB gene cluster (K02305, K04561) for the reduction of nitric oxide (NO) to nitrous oxide (N_2_O); however, only *M. szegediense* O-12 and Norfolk had the *nir*K (K00368) for dissimilatory nitrite reductase for denitrification, and none of them had the alternative *nir*S (K15864) nitrite reductase. Similarly to other methanotrophic species, homologs of hydroxylamine dehydrogenase (K10535) responsible for NH_2_OH oxidation to NO_2_, as well as hydroxylamine reductase reducing NH_2_OH to form NH_4_^+^ (K05601), are present in the three genomes. Additionally, the three strains have the genetic potential to assimilate NH_4_^+^ through the glutamate cycle, with their genes for glutamine synthetase (GS) *gln*A (K01915) and glutamate synthase (GOGAT) *glt*BD (K00265, K00266) [58]. The three strains have the genetic potential for the reversible conversion of glutamate to 2-oxoglutarate/α-ketoglutarate (which is an intermediate of the Krebs cycle) and ammonia through their glutamate dehydrogenase GDH2 (K15371). Additionally, the three strains also have the gene for alanine dehydrogenase *ald* (K00259), which had been described to participate in the reductive amination of pyruvate in other methanotrophs, under high NH_4_^+^ environments [58]. The two *M. szegediense* strains also have the gene for another glutamate dehydrogenase *gdh*A (K00262), which has been demonstrated to be required for *Streptococcus pneumoniae* for adaptation to high temperatures (40 °C) [61].

### 3.13. Carbon Storage

The carbon storage inventory includes glycogen biosynthesis *glg*ABC as well as the genes for polyhydroxybutyrate biosynthesis (Appendix A). The key genes for polyhydroxyalkanoate synthase *pha*C (K03821), as well as acetyl-CoA C-acetyltransferase *phaA* (K00626) and 3-oxoacyl-[acyl-carrier-protein] reductase *phaB* (K00023), were present in the three *Methylocaldum* strains. Furthermore, additional *pha*C homologs were found in the three strains, indicating possible PHB co-polymer biosynthesis. The ability to produce polyhydroxyalkanoates has been recently observed for axenic cultures of *Methylocaldum*, supporting the genetic observations [62]. In addition to genes for polymer biosynthesis, all three strains contain genes encoding pathways of sucrose biosynthesis.

### 3.14. Pyomelanin/HGA-Melanin Proposed Production

A signature characteristic of *Methylocaldum* strains is the development of light to dark brown-colored colonies/culture [63]. Previous analyses showed that *M. szegediense* O-12 synthesizes a tyrosine-derived melanin-like pigment upon a decrease in growth temperature (at a suboptimal temperature of 42 °C) [64]. Because of the lack of the canonical *mel*C1 for tyrosinase (K00505) for the production of eumelanin in *Methylocaldum*, here it is proposed that this color corresponds to a type of melanin, known as pyomelanin or HGA-melanin [65]. The three *Methylocaldum* strains have genes encoding the tyrosine degradation pathway via homogentisic acid (HGA), *hpp*D (K00457), and *hmg*A (K00451). In this pathway, the HGA intermediate can continue to be reincorporated into central metabolism as fumarate and acetoacetate [66] or accumulated and, through an spontaneous autoxidation process, converted to benzoquinone, which polymerizes to form pyomelanin [67,68] (Figure 7). The gene encoding histidinol-phosphateaminotransferase *his*C (K00817), which produces the HGA precursor 4-hydroxyphenylpyruvate, has four CDS assigned to it in *M. marinum* S8, three CDS in *M. szegediense* O-12 and two CDS in Norfolk.

## 4. Discussion

Here, three genomes of thermophilic/thermotolerant methanotrophs of the genus *Methylocaldum* were analyzed. Most methanotrophs display metabolic plasticity in response to the availability of key nutrients (nitrogen, phosphates) or metals (copper, tungsten, lanthanides). However, all three members of the *Methylocaldum* genus are superior to other comparable methanotrophs in the number of paralogs for key enzymes as well as the number of metabolic pathways for C_1_-carbon conversions and storage. The comparison revealed several key findings regarding their genetic diversity and metabolic potential.

All three strains possess multiple gene clusters for particulate methane monooxygenase (pMMO), suggesting redundancy in methane oxidation capabilities. This redundancy may provide metabolic flexibility, allowing *Methylocaldum* strains to thrive in diverse environmental conditions. The analysis of the genes involved in nitrogen metabolism revealed the presence of nitrogen fixation genes in all three strains, indicating their potential ability to fix atmospheric nitrogen. Additionally, genes involved in nitrate assimilation and nitric oxide reduction are present, suggesting versatility in nitrogen metabolism.

The genome comparison of the three *Methylocaldum* strains revealed insights into their genetic composition and metabolic potential. *M. szegediense* O-12 and Norfolk exhibit similar genomic characteristics, including genome size, GC content, and predicted protein-coding genes. However, *M. marinum* S8 diverges significantly, with a larger genome size and a lower percentage of orthologous genes shared with the *M. szegediense* strains. Phylogenetic analysis based on 16S rRNA sequences and ANI x AC confirmed the close relationship between *M. szegediense* O-12 and Norfolk, while *M. marinum* S8 formed a distinct clade. Genome synteny analysis highlighted variations in methane monooxygenase gene clusters among the strains, suggesting potential differences in methane oxidation pathways. Comparative abundance analysis revealed shared and unique functional categories, with genes related to energy production, metabolism, and transport being predominant. In terms of the genetic toolkit for methane oxidation, the *Methylocaldum* strains display redundancy in key enzymes involved in methane metabolism, including multiple copies of *pmo*C genes and methanol dehydrogenases. Additionally, differences in the presence of formate dehydrogenase genes suggest potential variations in formate utilization strategies among the strains. The genomes also encode pathways for single-carbon assimilation, including the ribulose monophosphate (RuMP), serine, and Calvin–Benson–Bassham (CBB) cycles. Nitrogen metabolism genes indicate the capacity for nitrogen fixation and assimilation, with differences in nitrate transport systems among the strains. Carbon storage pathways involve genes for glycogen and polyhydroxybutyrate biosynthesis, suggesting adaptations for carbon storage under varying environmental conditions. Unique functions identified in *M. marinum* S8 include catalase, involved in the detoxification of hydrogen peroxide, and a nearly complete de novo cobalamin biosynthetic pathway, enabling independent synthesis of the essential cofactor. These genomic features may confer advantages for survival and adaptation in specific ecological niches. Furthermore, *M. marinum* S8 harbors additional hydrogenase subunits and genes related to molybdopterin biosynthesis, indicating potential metabolic capabilities for hydrogen metabolism and cofactor biosynthesis. The presence of putative stress response genes and heavy metal-binding proteins suggests adaptation to diverse environmental conditions. O-antigen biosynthesis genes unique to the *M. szegediense* strains offer insights into cell envelope diversity and potential adaptation mechanisms. These differences may contribute to niche specialization and stress resistance in different environments. Lastly, the proposed production of pyomelanin/HGA-melanin in *Methylocaldum* strains suggests a potential mechanism for pigment production and adaptation to suboptimal growth conditions. Genes involved in tyrosine degradation pathways indicate the capacity for melanin synthesis via homogentisic acid accumulation. The significance of preservation of orthologs in organisms inhabiting very different ecological niches remains to be elucidated. However, it is tempting to propose that the evolution of C1 genes was driven by an adaptation to tolerate dramatic changes in temperature.

In summary, genomic analysis of *Methylocaldum* strains reveals extensive genetic diversity and metabolic versatility, with implications for their ecological roles and environmental adaptation strategies. Further research is needed to elucidate the functional significance of these genomic features and their contributions to the ecological success of *Methylocaldum* in diverse habitats.

## 5. Conclusions

Our research aimed to investigate the genomic characteristics and metabolic potential of well-physiologically characterized *Methylocaldum* strains. Here, we conducted a comparative genomic analysis of three strains: *M. szegediense* O-12 and Norfolk and *M. marinum* S8. Through phylogenetic analysis, genome synteny analysis, and comparative abundance analysis, we identified similarities and differences in their genetic composition and metabolic pathways, particularly focusing on methane oxidation, single-carbon assimilation, nitrogen metabolism, carbon storage, stress response, and pigment production. The results of our study hold broader significance beyond *Methylocaldum* for several reasons. Firstly, the genomic features and metabolic pathways identified in *Methylocaldum* strains provide insights into microbial ecology and biogeochemical cycling. A deeper understanding of the metabolic potential of *Methylocaldum* adds to the current collective knowledge of methanotrophic bacteria, which can be used for biotechnological applications, such as bioconversion and bioenergy production. Additionally, the genetic diversity and metabolic versatility observed in *Methylocaldum* may serve as a model for studying microbial adaptation and evolution in response to changing environmental conditions. Insights gained from studying *Methylocaldum* strains will support future efforts to unveil the ecology and biogeochemical implications of methanotrophic communities in diverse habitats, thus contributing to our broader understanding of microbial ecology and ecosystem functioning.

## Figures and Tables

**Figure 1 microorganisms-12-00469-f001:**
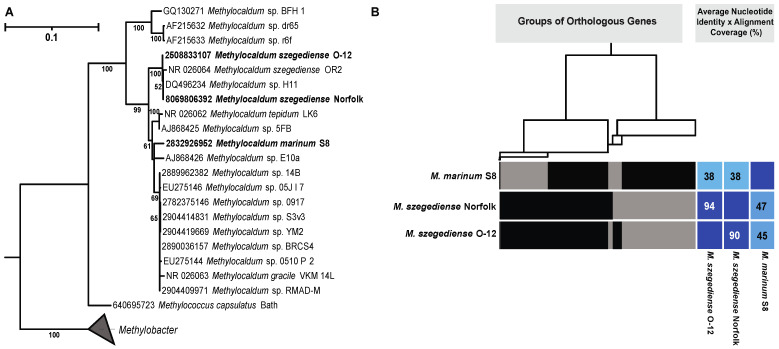
(**A**) Maximum likelihood tree representing the phylogenetic relationship of *Methylocaldum* based on 16S rRNA gene sequences. Leaf names with the strains analyzed in this article are in bold. Identifiers for each IMG gene ID are at the tip of each leaf. (**B**) Dendrogram showing the association between cluster of groups of orthologous genes, represented by thin vertical black lines, which, when grouped, form the rectangular blocks depicted on each genome. The clustering is based on the presence or absence of the orthologous genes on each genome. Average nucleotide identity (ANI) times the alignment coverage (AC) percentage of each genome when compared with one of the other *Methylocaldum* strains are shown inside of blue or light blue squares.

**Figure 2 microorganisms-12-00469-f002:**
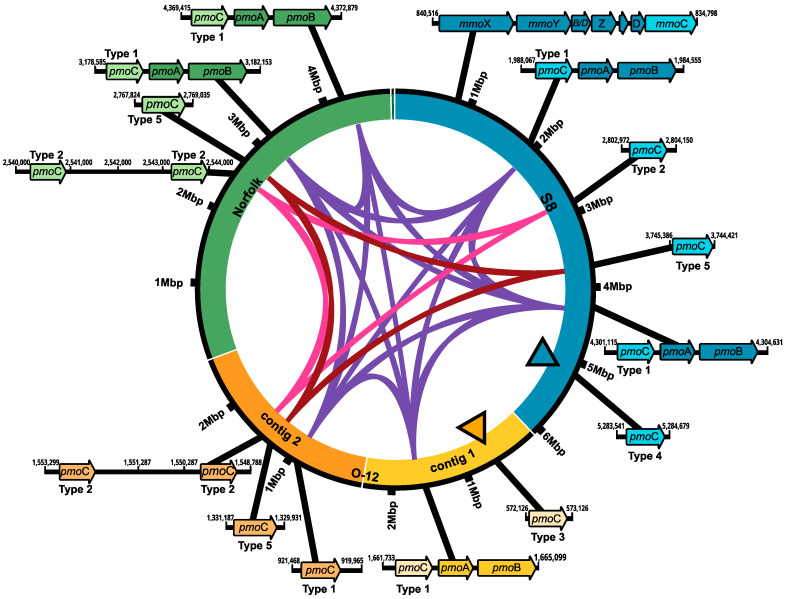
Synteny of methane monooxygenase (MMO) gene clusters in *Methylocaldum* genomes. Schematics for genomes of *M. marinum* S8, *M. szegediense* O-12, and *M. szegediense* Norfolk depicting the locus of MMO gene clusters. Ribbons connecting genomes show *pmo*C genes with high identity as part of pMMO gene cluster (purple) and as stand-alone genetic component (pink and magenta). Black-edge triangles indicate stand-alone copies of *pmo*C without high identity with any other *pmo*C within the three genomes. The gene cluster for soluble methane monooxygenase, sMMO, which is only found in *M. marinum* S8, is also shown.

**Figure 3 microorganisms-12-00469-f003:**
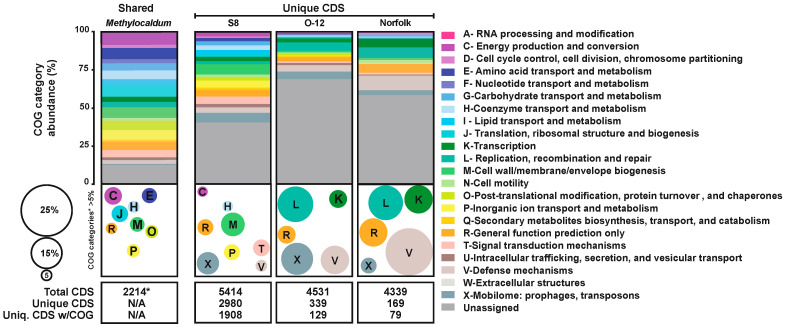
Comparative abundance of COG categories for *Methylocaldum* genomes. The relative percentage of each COG category is depicted by colored horizontal bars and bubbles according to the color-coded legend. Horizontal bars represent the percentage of coding sequences (CDS) assigned to each COG category among the three strains (shared) and the uniqueness of each strain (S8, O-12, and Norfolk). The total number of CDS considered for this comparison is at the bottom of each column. * indicates shared CDS; N/A indicates not applicable for the shared genes column. Bubble sizes and colors depict COG categories that represented >5% of the total when unassigned CDS were not considered.

**Figure 4 microorganisms-12-00469-f004:**
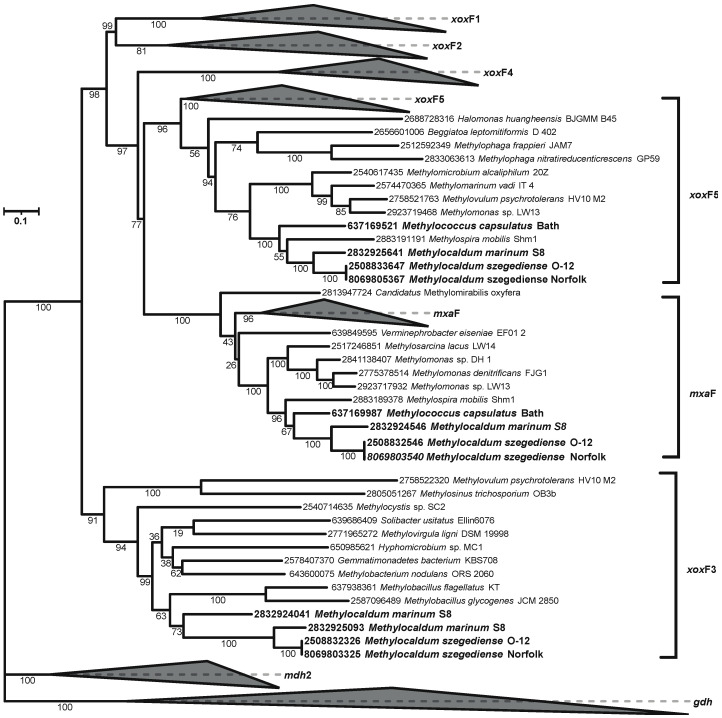
Phylogenetic reconstruction based on all *xox*F and *mxa*F genes found in the three *Methylocaldum* genomes analyzed (highlighted in bold). The corresponding IMG gene IDs are at the tip of each leaf. Collapsed clades did not contain any genes from the three analyzed genomes.

**Figure 5 microorganisms-12-00469-f005:**
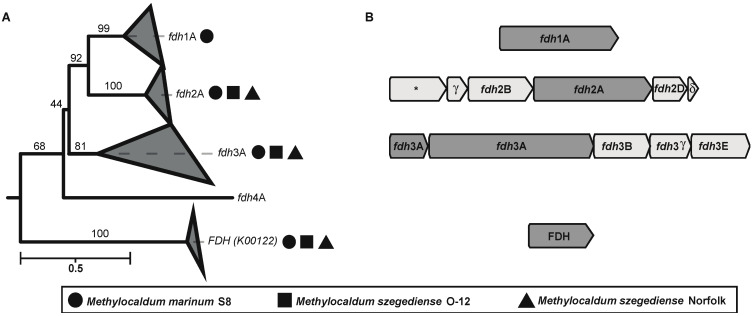
(**A**) Maximum likelihood tree reconstructing the phylogenetic relationship between the amino acidic sequences of four different types of formate dehydrogenase major subunits (K00123). Geometric figures next to each collapsed clade indicate the presence of a sequence from one of the three *Methylocaldum* strains analyzed in this study. Circles represent *M. marinum* S8, squares represent *M. szegediense* O-12, and triangles represent *M. szegediense* Norfolk. Formate dehydrogenase (FDH) K00122 was used as the outgroup. (**B**) Gene cluster genomic arrangement found on the three *Methylocaldum* strains analyzed in this study. The asterisk indicates the presence of a predicted protein with unknown function.

**Figure 6 microorganisms-12-00469-f006:**
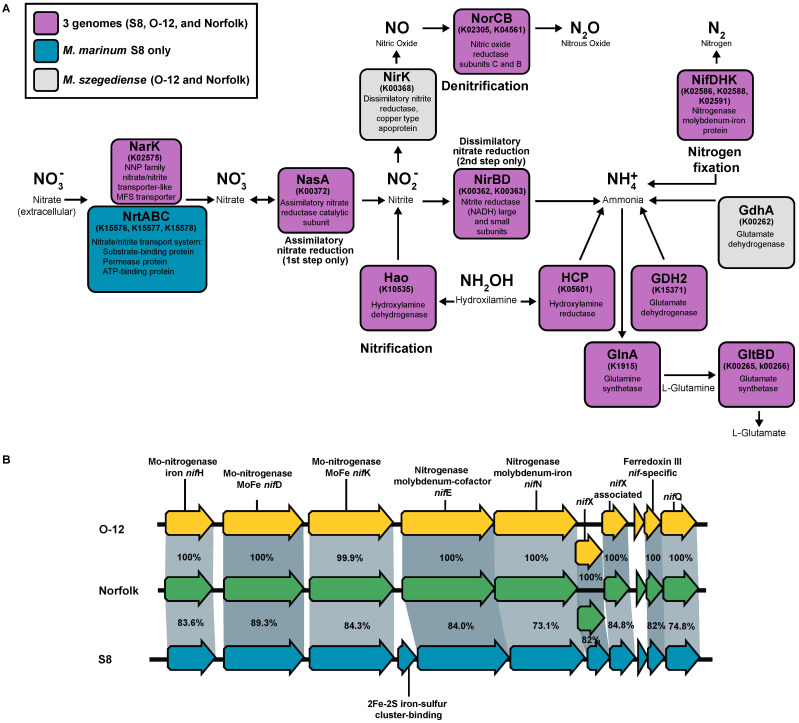
Nitrogen metabolism genes in *Methylocaldum* strains. (**A**) Diagram representing the presence and absence of genes involved in nitrogen cycling. (**B**) Synteny of nitrogenase gene cluster present in the three *Methylocaldum* strains. Pairwise identity percentage is indicated inside ribbons connecting homologous genes.

**Figure 7 microorganisms-12-00469-f007:**
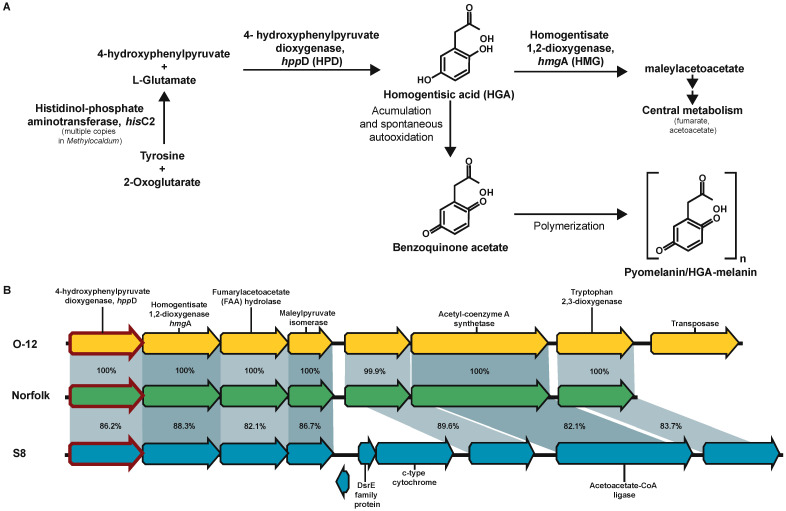
Proposed pyomelanin/HGA-melanin synthesis pathway via accumulation of homogentisic acid (HGA) in *Methylocaldum*. (**A**) Schematic representation of tyrosine degradation pathway having HGA as intermediate synthetized by 4-hydroxyphenylpyruvate dioxygenase (*hpp*D, HPD, K00457), which can be degraded via homogentisate 1,2-dioxygenase (*hmg*A, HMG, K00451) to be reincorporated into central metabolism as fumarate and acetoacetate or accumulated and converted to pyomelanin/HGA-melanin via polymerization of the monomer benzoquinone acetate. (**B**) Synteny of gene cluster where the *hpp*D gene (red-edge arrows) was present in the three *Methylocaldum* strains. Pairwise identity percentage is indicated inside ribbons connecting homologous genes.

## Data Availability

Data are contained within the article and Appendix A.

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
