# Peer review of "Genomic Insights into Moderately Thermophilic Methanotrophs of the Genus Methylocaldum"

_microorganisms, 2024, doi:10.3390/microorganisms12030469_

Round 1

Reviewer 1 Report

Comments and Suggestions for Authors In this papers, the authors study the pangenome of three strains, two
species of the genus Methylocaldum. The results of this study include,
one novel genome assembly, a genus-level, 16S-based phylogeny of the
genus, an analysis of the synteny of methane monooxygenase (MMO) gene
clusters between the three studied strains, and an indepth pangenome
analysis of the three studied strains. This study provides a good
analysis of three strains of methane-utilizing bacteria.

I found this paper to be generally technically sound. The organization
was straightforward and easy to follow. The writing was concise and
high quality (e.g., I found no grammatical errors or awkward
phases). I also felt that the value of the paper would have been
greater had the authors included a larger number of strains in their
analysis. I appreciate that collecting and sequencing more strains of
Methylocaldum might have been outside of the scope of the study, but
the authors did not include all of the publicly available genomes in
their study. Why?

Here is a list of all of the Methylocaldum genomes available in RefSeq
today:

| organism\_name | infraspecific\_name | isolate | assembly\_level | seq\_rel\_date |
|:-------------------------------|:--------------------|:------------|:----------------|:---------------|
| Methylocaldum szegediense O-12 | strain=O12 | na | Scaffold | 2013/09/05 |
| Methylocaldum sp. 14B | strain=14B | na | Contig | 2017/02/21 |
| Methylocaldum marinum | strain=S8 | na | Complete Genome | 2017/09/08 |
| Methylocaldum sp. RMAD-M | strain=RMAD-M | na | Contig | 2021/04/07 |
| Methylocaldum szegediense | na | Msz(Nor) | Complete Genome | 2023/03/31 |
| Methylocaldum sp. | na | SMAG_U12066 | Contig | 2024/01/14 |

The strains analyzed in the study were "O12", "S8", and
"Msz(Nor)". Why were "14B" and "RMAD-M" not included in this study?

Another issue I had with this manuscript was that in the introduction,
it was stated that the purpose as of the study was to "identify key
genes potentially involved in [the strains'] capacity to grow at
elevated temperatures". There are a handful of places in the Results
section where the authors return to this point, but it is barely
touched on in the Conclusions.

Which brings me to another problem that I have with the current
manuscript. Here is the Discussion ("Conclusions") sections in its entirety:

> Here we analyze three genomes of thermophilic/thermotolerant
> methanotrophs of the genus Methylocaldum. Most methanotrophs display
> metabolic plasticity in response to availability of key nutrients
> (nitrogen, phosphates) or metals (copper, tungsten,
> lanthanides). However, all three members of the Methylocaldum genus
> are superior to other microbes in the number of paralogs for key
> enzymes as well as the number of metabolic pathways for C1-carbon
> conversions and storage. The significance of preservation of
> orthologs in organisms inhabiting very different ecological niches
> remains to be elucidated. However, it is tempting to propose that
> the evolution of C1-genes was driven as an adaptation to tolerate
> dramatic changes in temperature.

That is only 104 words. The Results section is 10 pages, and the
Discussion is 104 words. To be fair to the authors, there is a fair
amount of discussion-level material embedded in the Results
section. This was done, I believe in order to put the details of the
results in context. However, the authors need to give more space at
the end of the paper to addressing, (a) to summarize how have they
addressed their self-identified research goal, and (b) why their
results matter beyond Methylocaldum.

In addition to the above, I have the following some minor criticisms:

Line 53, you need to include details of the sequencing and assembly of
the Norfolk strain, including DNA isolation and prep, details of the
sequencing (GridION + Illumina), the assembly pipeline (raw read
filtering? Unicycler). You need to provide the SRA accessions for the
raw reads as proof of data availability.

Line 53, taxon id's do not (generally) uniquely identify a specific
genome assembly. Please provide assembly accessions for all of the
genomes used in this study.

Line 66, please do not use website links in lieu of proper
citations. The link you use provides a list of appropriate citations
for MAFFT; please use the appropiate ones.

Line 66, version numbers should be provided for all software. Please
provide the version of MAFFT used.

Line 68, please provide version and citation for MEGA X.

Line 77, "circos" should be "Circos". Also, please provide version and
citation.

Line 80, "prodigal" should be "Prodigal". Also please provide version.

Line 83, FIXME [28] pangenome analysis

Line 87, "euclidean" should be "Euclidean". "ward" should be
"Ward". Also, there are multiple algorithms for Ward's method. Which
did you use? Hint, if you did this analysis in R, you can just name
the algorithm (e.g., "ward.D", "ward.D2", or so on). But if that's the
case, then you also need to provide a version and citation for R.

Line 91, "was generated using anvi-compute-genome-similarity to
obtain". This should be "a-c-g-s function" or "a-c-g-s workflow" or
whatever the appropriate terminology is for Anvi'o. Also, "a-c-g-s"
should be put in quotes or a different font to denote that it is a
software literal.

Line 92, "ANI X AF". According to the documentation
(https://anvio.org/help/main/programs/anvi-compute-genome-similarity/),
"a-c-g-s" computes ANI, not ANI X AF. Please clarify. Also, which of
"pyANI", "fastANI", or "sourmash" did you use?

Line 499, I cannot find Appendix A. I assume that the publisher did
not provide it to me. If it will not be available with the final
publication, then all references should be removed.

Author Response

We would like to thank all reviewers for their comments and insightful suggestions. In the revised version of the manuscript, we addressed the main concerns and modified our manuscript accordingly. The attached file includes point-by-point answers to the comments. 

Reviewer 2 Report

Comments and Suggestions for Authors

In this work, the authors performed the comparative genomic analysis of three microorganisms, including two thermophilic strains and one thermotolerant strain, in the genus Methylocaldum. In general, the research is important to understand how microorganisms adapt to or respond to elevated temperatures. However, the introduction is not clear, some results are not correct, and the discussion is not deep enough. I suggest to rewrite the manuscript to highlight the significance of the research.

Abstract

There are four species in the genus Methylocaldum, and the authors did not clearly explain why they selected the two species (the three strains). The different physiological features of the growth temperature, or different isolation niches.

After the analysis of this research, what is the main conclusion? Why do they have different characteristics of growth temperature from the genome information?

Why the comparative genomic analysis is necessary? The authors should express the significance of this work.

Introduction

The authors should introduce the optimal temperature for all of the three strains.

Results

L114: Do the chromosome and plasmid of Norfolk and the scaffold of S8 form circle DNA? In other words, are they complete?

L127: I do not think the phylogenetic analysis based only on 16S rRNA gene is necessary. A better choice is the concatenated amino acid sequences of conserved proteins, and it is named phylogenomic analysis.

L138: ANI boundary for species is about 95-96%, but not 90%. Or, other analyses should be provided, for example, AAI or dDDH.

I think O-12 and Norfolk are different species from Figure 1B.

L193: What is the difference between the thermophilic and thermotolerant species? Especially the function genes.

L430: We come back to the title and abstract, and we did not find any explanation why the authors selected the two species, one is thermophilic and the other one is thermotolerant. The results are so descriptive that I could not follow why the authors compare these genes.

L481: The accession numbers of the genomes should be provided the Data Availability Statement.

Author Response

We would like to thank all reviewers for their comments and insightful suggestions. In the revised version of the manuscript, we addressed the main concerns and modified our manuscript accordingly. Below are point-by-point answers to the comments. 

Round 2

Reviewer 2 Report

Comments and Suggestions for Authors

The authors have addressed my concerns. I suggest to accept the manuscript in present form.